# A Personalized Learning Path Recommendation Method Incorporating Multi-Algorithm

Yongjuan Ma [1], Lei Wang [1,2,*], Jiating Zhang [1], Fengjuan Liu [2] and Qiaoyong Jiang [1]

[1] The Key Laboratory of Network Computing and Security Technology of Shanxi Province, Xi'an University of Technology, Xi'an 710048, China; mayongjuan202306@163.com (Y.M.); zjt_fight@163.com (J.Z.); jiangqiaoyong@xaut.edu.cn (Q.J.)

[2] The Key Laboratory of Industrial Automation of Shaanxi Province, Shaanxi University of Technology, Hanzhong 723001, China; liufj06@126.com

\* Correspondence: wanglei_sut@163.com

**Abstract:** In this era of intelligence, the learning methods of learners have substantially changed. Many learners choose to learn through online education platforms. Although learners may enjoy more high-quality educational resources, when they are faced with an abundance of resource information, they are prone to become lost in knowledge, among other problems. To solve this problem, a multi-algorithm collaborative, personalized, learning path recommendation model is proposed to provide learning guidance for learners of online learning platforms. First, the learner model is constructed from four perspectives: cognitive level, learning ability, learning style, and learning intensity. Second, the association rule algorithm is employed to generate a sequence of knowledge points and to plan the learning sequence of knowledge points for learners. Last, the swarm intelligence algorithm is utilized to ensure that each knowledge point is matched with personalized learning resources with a higher degree of adaptability so that learners can learn using a more targeted approach. The experimental results show that the research results of this paper can, to a certain extent, recommend ideal learning paths to target users, effectively improve the accuracy of recommended resources, and thus improve the learning quality and learning effect of users.

**Keywords:** learner model; association rule algorithm; swarm intelligence algorithm; personalized recommendation; learning path generation

## 1. Introduction

With the rapid development of online education, the number of teaching resources on the internet also presents a rapidly growing trend. Many learners choose to acquire knowledge through online learning platforms. However, when faced with learning resources with a large amount of data, strong professionalism, and a complex knowledge structure, learners are prone to problems such as information overload and learning treks in the learning process, which makes it difficult for learners to apply these resources to build their own learning paths. Therefore, it is necessary to provide personalized learning path recommendation services for learners and to guide them to learn in a more efficient manner.

Personalized learning path recommendations can effectively integrate high-quality learning resources, optimize the allocation of learning resources, and give them a maximum role. When learners can better carry out personalized learning activities using high-quality resources, their academic performance can significantly improve without spending much time collecting and organizing. In recent years, many researchers have carried out a series of studies on the problem of individualized learning paths. Zhang et al. [1] constructed a multi-dimensional curriculum knowledge graph (MCCKG) in the computer field and proposed a method based on graph convolutional networks to model high-order correlation relationships on the knowledge graph, in order to more accurately capture learners' preferences and recommend the best learning path for learners. Sun et al. [2] proposed

a method to personalize the recommendation of English learning based on knowledge graphs and graph convolutional networks. Firstly, we construct a knowledge graph containing a large number of Junior High School English exercises, which are classified by the knowledge points. Secondly, a graph convolutional neural network is used to generate a personalized knowledge graph for each student. Finally, it provides students with in-depth personalized services by generating personalized learning paths. Son et al. [3] proposed a multi-objective optimization model as a knowledge-based recommender. This model can generate an appropriate learning path for learners based on their background and job goals. Cai et al. [4] proposed a knowledge demand model based on knowledge tracking. According to the historical learning track of learners, knowledge tracking is used to simulate the learning situation of learners and generate adaptive learning paths for each learner. Li and Zhang [5] proposed a learning path generation algorithm based on network embedding and learning effect, which uses a network embedding algorithm to learn the similarities among learners and then generates an adaptive learning path for target learners according to the similarity between all historical learners and target learners and the learning effect of historical learners. Cheng et al. [6] proposed a learning path recommendation scheme based on ontology, which uses a professional learning sequence as the basis to generate personalized learning paths. Since learners' mastery of knowledge is dynamic, the researcher provides personalized course learning paths for learners by simulating their actual knowledge growth process. Chen et al. [7] proposed a personalized learning path recommendation system based on LINE Bot. This model uses long short-term memory (LSTM) to construct video viewing preference characteristics, student clusters, and learning paths to recommend a personalized learning path that is suitable for each student. Its related recommendation content and predicted results are received by users via timely LINE messages to achieve the purpose of timely and active recommendation.

Starting from the three dimensions of learners, knowledge points, and resources, this paper designs a personalized learning path recommendation model with multi-algorithm collaboration, which solves the problem of learners' knowledge treks in massive learning resources and verifies the effectiveness of this model through a series of experimental designs, implementation, and data analysis. The experimental results show that, to some extent, the model designed in this study can recommend a more ideal learning path to learners, effectively improve the accuracy of recommended resources, and thus improve the learning quality and effect of learners.

The specific organization of this paper is as follows: Section 1 is the introduction section, which mainly describes the current status of research on personalized learning path recommendation. Section 2 is the problem description, which analyzes the problems of existing research and proposes the research content of this paper. Section 3 is the core methodology, which introduces the implementation process of the model in detail. Section 4 is the experimental analysis, which verifies the effectiveness and feasibility of the model proposed in this paper. Section 5 is the conclusion, which mainly describes the current research results and the details of future work.

## 2. Problem Formulation

The advent of educational informatization has promoted the rapid development of the education industry. Learners' learning methods are no longer limited to the traditional offline classroom, and learners can choose to use the internet for learning. Although online learning has numerous learning resources, it still requires learners to explore different knowledge points and related resources and to organize them. In this process, learners are prone to become lost in knowledge, which affects their learning efficiency. Therefore, how to identify suitable resources for learners among the massive amount of learning resources and plan good learning paths for learners so that they can learn more efficiently and in a more targeted manner. In this paper, a multi-algorithm collaborative personalized learning path recommendation model is designed to solve this problem. First, the cognitive level of learners is determined using the cognitive diagnosis model to identify the knowledge

points of learners. Then, the knowledge point sequence of learners is established by combining it with a knowledge graph. Second, to make the recommended learning path more personalized, it is necessary to analyse the characteristics of each learner. This paper describes the characteristics of learners from the four perspectives of cognitive level, learning ability, learning style, and learning intensity and establishes a learner model. Third, since learning resources are the carriers of knowledge points, this paper mainly establishes a personalized learning path recommendation model from two aspects: the matching degree between knowledge points and learning resources and the fitness between learners and resources. Last, this paper uses a swarm intelligence algorithm to solve the personalized learning path recommendation method and verifies the effectiveness of the particle swarm optimization (PSO) algorithm in the application of personalized learning path recommendation through simulation experiments.

## 3. Methodology

The recommendation of personalized learning paths uses learners as the centre and recommends appropriate learning paths for them according to their own characteristics (cognitive level, learning ability, learning style, and learning intensity) to improve learning efficiency. Therefore, in the online learning environment, the recommended learning path for learners should consider not only whether the sequence itself is suitable for learners but also the learning order of the target knowledge points. This paper proposes a learning path recommendation model based on multi-algorithm fusion to provide personalized services for learners in online learning environments. As shown in Figure 1, this model is mainly divided into two modules. The objective of the first module is to accurately generate the learning sequence from the target knowledge point for learners. After some processing, the data are input into the Fuzzy-CDF model to obtain the cognitive level of learners. Through the set threshold value $\mu$, the knowledge set that is not mastered is obtained. Then, the apriori association rule algorithm and the constructed knowledge map are employed to generate the knowledge sequence. The objective of the second module is to recommend personalized learning paths for learners (the combination of knowledge points and corresponding resources). Supported by the learning resource database and learner feature database, a swarm intelligence algorithm is applied to generate the final personalized learning path.

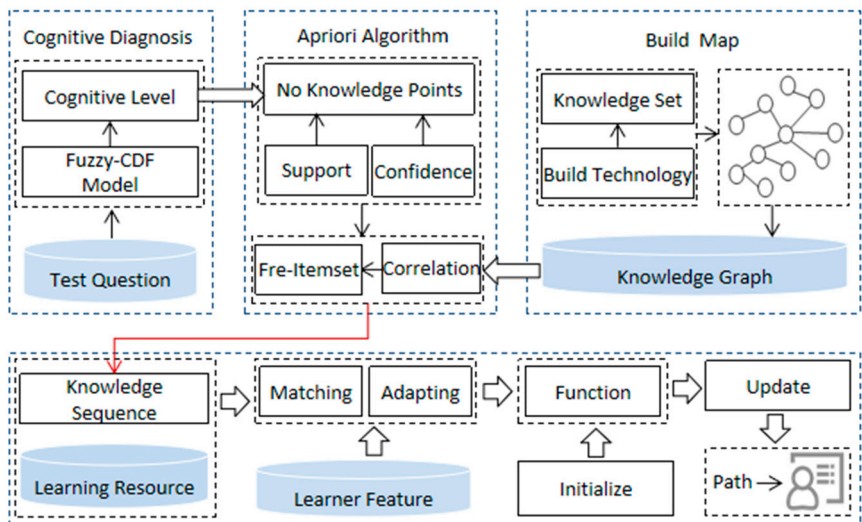

**Figure 1.** Framework of personalized learning path recommendation.

### 3.1. Related Work

In the era of intelligent education, building a personalized learner model is the key to changing the service mode of intelligent education, solving the problem of educational fairness, and achieving differentiated teaching. In the current research, the analysis of

learning behavior data is too single, making it difficult to build a more comprehensive learner model, which will make the personalized learning services provided have certain limitations. Therefore, this article delves into learning behavior data and constructs learner models from multiple perspectives, providing various support for personalized teaching and intelligent management. Collect learning behavior data from online learning platforms, and process and save this data to provide data support for constructing learner models. This article describes the feature information of learners from four perspectives: cognitive level, learning ability, learning style, and learning intensity, establishes a learner model, and provides a basis for subsequent personalized learning path recommendation methods.

3.1.1. Cognitive Level Based on Fuzzy-CDF

The cognitive level refers to the mastery of knowledge points, including the knowledge that is mastered and the knowledge that is not mastered. In cognitive psychology, the modelling process of learners' mastery of knowledge points is referred to as the cognitive diagnostic model [8]. In terms of personalized learning, typical cognitive diagnostic models include the DINA [9] model and the DINO [10] model, which mainly reflect learners' understanding of the knowledge points that they have learned and explain whether learners have mastered the current knowledge points. However, these models are secondary graded cognitive diagnostic models, namely, they only have two states: mastering and not mastering. To obtain the cognitive level of learners using a more fine-grained approach, this paper adopts a fuzzy cognitive diagnosis model, Fuzzy-CDF [11], to obtain learners' knowledge mastery. The output of this model is a continuous value between 0 and 1, which can more accurately obtain the learners' mastery of knowledge points. The Fuzzy-CDF model starts from the learner's potential characteristics, determines the learner's skill proficiency, calculates the examinee's mastery of the problem, and generates the examinee's observable score of the problem by considering the error and guess factors.

Fuzzy-CDF assumes that the set of students who master all (part) of the knowledge and skills required for Problem *i* is the intersection (union) of fuzzy sets related to skills [12]. Fuzzy-CDF uses fuzzy numbers to quantify subjective, qualitative, and uncertain information. It is assumed that students' knowledge ability level is their membership degree in the fuzzy set corresponding to the knowledge ability. Therefore, in objective questions, the mastery degree $\eta_{ji}$ of learner *j* on test question *i* is the fuzzy intersection of learner *j*'s cognition of knowledge points. In subjective questions, $\eta_{ji}$ is the fuzzy sum of learners' cognition of knowledge points. The specific calculation is presented as follows:

$$\eta_{ji} = \bigcap_{1 \le k \le K, q_{ik}=1} \mu_k(j) \tag{1}$$

$$\eta_{ji} = \bigcup_{1 \le k \le K, q_{ik}=1} \mu_k(j) \tag{2}$$

where $q_{ik} = 1$ indicates whether test question *i* has examined knowledge point *k* and *K* indicates the total number of knowledge points examined in test question *i*. Whether a learner correctly answers a question is not only related to the extent to which the learner has mastered the knowledge points investigated in the test but also may be affected by the factors of guess and error, resulting in the inability to accurately obtain the learner's true ability. Therefore, we can use the 4-parameter logistic model to estimate the error and guess parameters, and we can intervene in the acquisition of learners' abilities. The specific calculation is presented as follows:

$$P(X_{ij} = 1 | \theta_i; a_j, b_j, c_j, d_j) = c_j + (d_j - c_j) \frac{e^{1.7a_j(\theta_i - b_j)}}{1 + e^{1.7a_j(\theta_i - b_j)}} \tag{3}$$

$a_j$, $b_j$, $c_j$ and $d_j$ are the discrimination, difficulty, guess, and error parameters, respectively. The real scores of the learners on the objective and subjective questions are obtained. The specific calculation is presented as follows:

$$P(R_{ji} = 1 | \eta_{ji}, s_i, g_i) = (1 - s_i)\eta_{ji} + g_i(1 - \eta_{ji}) \tag{4}$$

$$P(R_{ji} = 0 | \eta_{ji}, s_i, g_i) = N(R_{ji} | (1 - s_i)\eta_{ji} + g_i(1 - \eta_{ji}), \sigma^2) \tag{5}$$

where $R_{ji}$ indicates learner $j$'s score on test Item $i$ and $s_i$ and $g_i$ indicate learner $j$'s mistakes and guesses, respectively, on test Item $i$.

### 3.1.2. Quantitative Learning Ability

Ability is developed and cultivated in the process of human activities and is a personalized psychological feature that plays a role in regulating and stabilizing the whole process. Learning ability is the psychological condition for learners to successfully complete learning tasks, acquire knowledge and achieve learning goals. In traditional educational activities, few researchers fully consider the impact of user learning ability differences and the combination of user information and behaviour [13]. Presently, in the field of smart education, researchers have gradually conducted relevant research on learners' learning ability. Yi et al. [14] evaluated learners' learning ability from two aspects: learning motivation and cognitive ability.

On the basis of the above research, this paper proposes the learning ability $S$, which is suitable for the learning path recommendation model. The learning ability of a learner reflects the comprehensive ability of learners in all aspects. Therefore, this paper constructs the learning ability of learners from four aspects: executive ability ($S^1$), adaptive ability ($S^2$), knowledge management ability ($S^3$) and ability to analyse and solve problems ($S^4$). The learning ability $S_i$ of learner $i$ is expressed as $S_i = (S^1 + S^2 + S^3 + S^4)/4$, where $S$ is a direct number between 0 and 1, $S_i \in [0, 0.3)$ is weak, $S_i \in [0.3, 0.7)$ is medium, and $S_i \in [0.7, 1]$ is strong.

1.  Executive Ability

Executive ability refers to whether learners can complete the given learning task within the specified time. Its value will affect learners' mastery of a certain knowledge point, thus affecting learners' academic performance. This paper mainly quantifies executive ability and calculates the completion degree of learners' learning tasks. The specific calculation is presented as follows:

$$S_i^1 = \frac{\sum_{j=1}^{n} \left( \frac{\sum_{k=1}^{m} z_{jk}}{m} \right)}{n} \tag{6}$$

where $S_i^1$ indicates the execution ability of learner $i$, $z_{jk}$ indicates the completion of learning Item $k$ corresponding to learning task $j$, $n$ indicates the number of tasks that learners need to complete, and $m$ indicates the number of learning items corresponding to each learning task. Thus, $S_i^1$ takes the average value of all learning tasks completed by learners.

2.  Adaptive Ability

In the era of intelligent education, learners can achieve better learning results only if they efficiently and quickly adapt to the learning environment. In this paper, the quantification of learners' adaptability mainly considers the aspects of learners' concentration and interaction. Concentration is quantified by the total login time and total leave time

of learners on the platform. The degree of interaction mainly reflects the interaction of learners on the platform. The specific calculation is presented as follows:

$$S_i^2 = \alpha \sum_{j=1}^{n} \frac{T_j}{\sum_{k=1}^{m} t_{jk}} + (1-\alpha)\left(\frac{C_f + C_p + C_x + C_d + C_s}{5}\right) \tag{7}$$

where $S_i^2$ indicates the learner's adaptive ability; $T_j$ indicates the total login time of learner $i$ in the $j$th login; $t_{jk}$ indicates the total leave time of learner $i$ in the $j$th login; $n$ indicates the total login times; $m$ indicates the total leave times each time; $C_f$, $C_p$, $C_x$, $C_d$, and $C_s$ indicate the number of posts, comments, downloads, likes, and collections, respectively, of learners on the platform; is the weight coefficients, with a value of 0.6.

3. Knowledge Management Ability

Knowledge management ability reflects the learners' ability to integrate, consolidate and memorize learned knowledge points to avoid the loss of knowledge points, which is reflected in the learners' upload rate of notes, the completion rate of exercise questions, and Q & A. This ability is very important in the whole learning process and enables learners to reunderstand what they have learned. This paper mainly quantifies knowledge management ability and calculates the rate of note uploading, the rate of completing exercises, and the rate of answering questions. The specific calculation is presented as follows:

$$S_i^3 = \frac{1}{1 + e^{-(x_i - \overline{x})}} \tag{8}$$

where $S_i^3$ indicates the knowledge management ability of learner $i$, $\overline{x}$ indicates the average value of all learners' note upload rates, exercise completion rates, and question answering rates, and $x_i$ actually denotes the amount of learner $i$ management behaviour on the learned knowledge points. Its value is jointly quantified by the note upload rate ($u$), exercise completion rate ($w$), and question-answering rate ($d$). The specific calculation is presented as follows:

$$x_i = \frac{\sum_{j=1}^{N} u_{ij}}{m} + \frac{\sum_{j=1}^{N} a_{ij}}{n} + \frac{\sum_{j=1}^{N} v_{ij}}{q} \tag{9}$$

$m$, $n$, and $q$ indicate the total number of notes, exercises, and Q & A, respectively; $u_{ij}$, $a_{ij}$, and $v_{ij}$ indicate the number of uploaded notes, completed exercises, and Q & A of learner $i$ for knowledge point $j$, respectively; and $N$ indicates the total number of knowledge points.

4. Ability to analyse and solve problems

The ability to analyse and solve problems is a comprehensive ability of both analytical ability and solving ability. This ability tests the application of learners' existing knowledge, that is, learners use their own knowledge and skills to analyse and solve problems. This ability reflects whether a learner has a thorough grasp of the learned knowledge and is an important way to promote learners' cognitive level. On the basis of the learners' existing cognitive level, this paper quantifies the accuracy of the learners' 'answers. The specific calculation is presented as follows:

$$S_i^4 = c_0 + \frac{(1 - c_0)}{1 + e^{-\left(\frac{\sum_{j=1}^{n} c_{ij}}{n} - \overline{c}\right)}} \tag{10}$$

where $S_i^4$ indicates the learner's ability to analyse and solve problems, $c_0$ indicates the learner's initial cognitive level, $c$ indicates the average of all learners' correct rates, $n$ indicates the total number of test questions, $c_{ij}$ indicates the learner's correct rate for test question $j$, that is $c_{ij} = cor_{ij}/m$, $m$ indicates the total number of questions for a set of test questions, and $cor_{ij}$ indicates the number of questions that learner $i$ answers correctly on test question $j$.

### 3.1.3. Learning Style Based on the Felder Silverman Scale

Learning style refers to the notion that in the learning process, each learner shows different learning tendencies. To provide learners with more suitable and attractive learning content, the learning style of learners can be modelled to enable all learners to learn according to their own learning style, thus realizing the personalization of online learning. According to the Felder Silverman learning style scale, learning styles are classified into four dimensions, and quantified by eight different types, namely, information processing: active and contemplative; perceptual information: perceptive and intuitive; information input visual and verbal; and content understanding: sequential and comprehensive [15]. Based on the Felder Silverman learning style scale and the research content, this paper proposes a behaviour model and threshold suitable for this learning style, as shown in the Table 1 below.

**Table 1.** Learning style quantification table.

| Learning Style | B-P | Pattern Quantization | Threshold | |
|---|---|---|---|---|
| | | | L<->M | M<->H |
| Active/Meditating | P-V (+) | Posting data of all learning tasks | <2 | >5 |
| | PB-V (−) | Reply data of all learning tasks | <10 | >30 |
| | FV-T (+) | (Forum visit/total course) × 100% | <5% | >15% |
| | D-A (+) | Discussion amount of learning task | <2 | >5 |
| | T-A (+) | Test amount of learned task | <2 | >5 |
| Perceptual/Intuitive | DV-A (+) | (Document views/total views) × 100% | <50% | >75% |
| | DB-T (+) | (Relative brows/total brows) × 100% | <75% | >100% |
| | DD-T (−) | (material downloads/total downloads) × 100%—Video, courseware, exercises | <50% | >75% |
| | TD-T (+) | (Test-S/maximum-S allowed) × 100% | <70% | >90% |
| Visual/Verbal | VB-T (+) | (Actual view-T/expected view-T) × 100% | <75% | >100% |
| | VP-A (+) | (Video playback-S/total playback-S) × 100% | <75% | >100% |
| | MV-A (+) | (Map views/total views) × 100% | <50% | >75% |
| | MB-T (−) | (Actual browsing duration of the atlas/total browsing duration) × 100% | <75% | >100% |
| Sequential/Comprehensive | CC-A (+) | (Times of clicking chapter button/total times) × 100% | <30% | >70% |
| | KC-A (−) | (Number of clicks on knowledge points/total times) × 100% | <30% | >70% |
| | S-A (+) | (Number of search maps/total search times) × 100%—comprehensive maps, sub maps of chapters | <30% | >70% |

In the table, "+" and "−" represent the behaviour mode on the left and the behaviour mode on the right, respectively, that most affect each group's learning style dimension. For example, the more posts that you post, the more you can reflect that the learners' learning style belongs to the left active type, and the fewer replies that you have, the more you can reflect that the learners' learning style belongs to the right meditative type. The learning style $C$ of each learner is affected by a variety of behaviour modes $P_1, P_2, \ldots, P_n$, so it is stipulated that the threshold of quantifying the learner's behaviour mode is divided within the L m range, then $P_i \in \{H\}$; if it is within the M-H range, then $P_i \in \{L\}$; if it is in

other ranges, then $P_i \in \{M\}$. In summary, a certain behaviour mode $P_i$ of learner $j$ can be described as $P_i^j$. According to the description, $P_i^j$ is defined as:

$$P_i^j = \begin{cases} 1, P_i^j = H \\ 0, P_i^j = M \\ -1, P_i^j = L \end{cases} \tag{11}$$

then, the learning style $C$ of learner $j$ in the number of $P_{count}$ behaviour modes is calculated by the following formula:

$$V_j(C) = \frac{\sum\limits_{i=1}^{n} P_i^j}{n} \tag{12}$$

3.1.4. Learning Intensity Modelling

Learning intensity refers to the energy that learners put into completing learning activities on the platform. This paper describes the learning intensity of learners from three aspects, namely, learning engagement, participation, and activity.

- Learning Engagement

The learning input of learners in the platform is measured according to the learning time of learners in resources, exercises, and video pages. The specific calculation is presented as follows:

$$c_i = \frac{t_i - miT}{maT - miT} \tag{13}$$

where $c_i$ indicates learning engagement, $t_i$ indicates the learners' input duration on Page $i$, $maT$ indicates the upper limit of input duration, and $miT$ indicates the lower limit of input duration. According to the analysis of learners' learning duration data in resources, exercises, and video pages, learners' learning duration is less than 300 min, and those with a learning duration of less than 20 min are not considered. Therefore, $maT = 300$ and $miT = 20$ in this paper.

- Participation

Participation refers to the amount of learners' behaviour in completing learning activities on the platform. Its quantitative value p is measured by the number of tests ($T$), number of homework submissions ($H$), and number of discussions ($D$). The specific calculation is presented as follows:

$$p = \eta_1 T + \eta_2 H + \eta_3 D \tag{14}$$

where $\eta_1 + \eta_2 + \eta_3 = 1$, $\eta_1 = 0.5$, $\eta_2 = 0.3$, and $\eta_3 = 0.2$, indicating the degree of influence of the three behaviours on learner participation. To ensure the high quality of the number of learners' tests, submitted assignments, and discussions, $T$, $H$, and $D$ are set to $[0, \sigma + \mu]$ with inner linear growth; $[\sigma + \mu, +\infty]$ decreases with a trend of $1/x$, where $\mu$ is the mean value within the respective values of $T$, $H$, and $D$; and $\sigma$ is the standard deviation.

- Activity

Activity refers to the amount of interaction of learners on the platform, and a is the quantitative value of the number of posts ($S$), number of forum replies ($C$), number of resource sharing ($R$), and number of likes ($L$). The specific calculation is presented as follows:

$$a = \eta_1 S + \eta_2 C + \eta_3 L + \eta_4 R \tag{15}$$

where $\eta_1 + \eta_2 + \eta_3 + \eta_4 = 1$, $\eta_1 = 0.4$, $\eta_2 = 0.3$, $\eta_3 = 0.2$, and $\eta_4 = 0.1$ to ensure a high-quality number of posts, forum replies, resource sharing, and likes; $S$, $C$, $L$, and $R$ are at $[0, \sigma + \mu]$ with inner linear growth; $[\sigma + \mu, +\infty]$ decreases with a trend of $1/x$, where

$\mu$ is the mean value within the respective values of *S*, *C*, *L*, and *R*; and $\sigma$ is the standard deviation. The specific calculation is presented as follows:

$$L_i = c + \frac{1}{1 + e^{-(p - \overline{p})(a - \overline{a})}} \tag{16}$$

where, *c* represents learning engagement, which is measured based on learners' learning time in resources, exercises, and video pages. The specific calculation is shown in Equation (13) above. $\overline{p}$ and $\overline{a}$ are the mean values of participation and activity, respectively, of all learners. $L_i \in [0, 0.5)$ indicates weak learning intensity, $L_i \in [0.5, 0.7)$ indicates medium learning intensity, and $L_i \in [0.7, 1]$ indicates strong learning intensity.

### 3.2. Generating Knowledge Point Sequences

The apriori algorithm is one of the classical algorithms of association rules. In 1993, Agrawal et al. proposed a basic algorithm for identifying frequent item sets. The principle of the apriori algorithm is to use an iterative method called layer-by-layer search, where the k item set is used to explore the k + 1 item set [16]. In this paper, we use the apriori algorithm to mine the learning results of learners on an online learning platform, to determine the frequent item sets of knowledge points that learners do not master, and then to combine an existing knowledge map library to reasonably plan the sequence of knowledge points suitable for learners.

The algorithm mainly obtains the frequent patterns we need from a huge dataset according to the given minimum support and confidence. Based on frequently received criteria, links between individual data can be found. The layer-by-layer iterative method was adopted by Apriori [17]. The generation of knowledge point sequences is supported by learners' existing cognitive levels and knowledge map databases. The set $kg = \{k_1, k_2, \ldots, k_n\}$ is the set of *n* knowledge points in the knowledge map library, and the points are arranged according to the relationship between knowledge points. Set $cg = \{c_1, c_2, \ldots, c_n\}$ contains learners' cognitive levels, specifically obtained by the Fuzzy-CDF model according to the learners' test question library. To obtain a set of knowledge points that learners do not master, the threshold $\mu_{cog}$ is set, and the cognitive level is lower than the $\mu_{cog}$ value, which refers to the knowledge points that are not mastered. The set $ng = \{n_1, n_2, \ldots, n_m\}$ represents the set of knowledge points that are not mastered, and $0 \leq m \leq n, ng \subset kg$. For learners whose target knowledge point is *B*, if they want to obtain knowledge point *A* and its association rule $A \Rightarrow B$, that is, if learners do not master knowledge point *A*, they will not master the target knowledge point, where $A \subset ng$, $B \subset ng$ or $B \subset kg$, but $A \cap B = \varnothing$. According to the principle of the apriori algorithm, frequent item sets need to be obtained according to support and confidence where support indicates the proportion of the dataset containing the current item set, and confidence indicates the probability of the occurrence of one item set when another item set occurs. If an item set meets the minimum support threshold, it is a frequent item set. If the item set also meets the minimum confidence threshold, it has strong association rules. Junior high school mathematics is used as an example to build a knowledge map of 584 knowledge points in the discipline. For any learner $L_i$ on the online learning platform, some answer records of the learner in the test question library are shown in Table 2. The threshold of cognitive level $\mu_{cog} = 0.9$. Those in the *cg* set that are lower than this threshold are knowledge points that are not mastered. The minimum support is minsupport = 0.53, and the minimum confidence is minconfindence = 0.91. The obtained frequent item set of knowledge points that are not mastered is shown in Table 3.

**Table 2.** The uncontrolled knowledge.

| Question No. | Knowledge Points Not Mastered (ng) |
|---|---|
| EX-NO-01 | 2, 4, 15, 23 |
| EX-NO-02 | 2, 7, 8, 10 |
| EX-NO-03 | 1, 5, 9, 20, 21 |
| EX-NO-04 | 19, 16, 3, 6 |
| EX-NO-05 | 11, 25, 14, 17 |

**Table 3.** Frequent item set of knowledge points.

| Serial No. | Itemset | Support (%) |
|---|---|---|
| FQ-IT-1 | 1, 4, 13, 17, 20 | 30 |
| FQ-IT-2 | 1, 2, 4, 7, 19 | 55 |
| FQ-IT-3 | 1, 4, 7, 17, 19 | 66 |
| FQ-IT-4 | 4, 7, 8, 11, 15 | 12 |
| FQ-IT-5 | 2, 3, 10, 21, 25 | 53 |

We obtain that the frequent item set of learner $L_i$ is $fq = \{1, 2, 3, 4, 7, 10, 17, 19, 21, 25\}$. When learners input target knowledge points, it is necessary to generate a sequence for learners to reach target knowledge points, that is, the learning sequence from $fq$ to target knowledge $n$ points. In the knowledge map, there is a certain dependency between two knowledge points. Therefore, this paper introduces a correlation to improve the accuracy of the knowledge point sequence. In the process of searching the current knowledge point to the target knowledge point, the correlation between the knowledge point and the precursor knowledge point is calculated for each encountered knowledge point, the knowledge point with the highest correlation is selected to continue the search, and the sequence of learners' knowledge points are obtained. The correlation calculation is presented as follows:

$$R_{\text{corr}}(v) = (1 - \theta) + \theta \times \sum_{u_i \in in(v)} \frac{sim_{uv} \times R_{corr}(u)}{\sum\limits_{u_k \in out(u)}} \tag{17}$$

where $R_{corr}(u)$ and $R_{corr}(v)$ represent the correlation between knowledge points $u$ and $v$; $\theta$ is the correlation coefficient, whose value is 0.9; $u_i \in in(v)$ is the number of edges connecting $u$ and $v$; $u_k \in out(u)$ is the number of all knowledge points (edges) connected with $u$; and $sim_{uv}$ is the similarity between knowledge points $u$ and $v$. This paper uses cosine similarity for the calculation.

### 3.3. Design of Recommendation Algorithm

In online learning systems, resources are the carrier of knowledge, and the learning path investigated in this paper is the combination of knowledge points and resources. In the previous section, a learning order (knowledge point sequence) of learners was obtained by the association rule algorithm, while in this section, learning resources are mainly matched for each knowledge point by the group intelligence algorithm according to the learning order of learners. To make the recommended learning path vary from person to person, this study also incorporates the characteristics of the learners themselves.

PSO is a well-established optimization algorithm that iteratively tries to find the global optimum, the original algorithm uses a population of particles that move around the search space according to the PSO update rules, each particle's movement is affected by its best-known position, and the global best-known position [18]. Therefore, in this paper, the PSO algorithm is employed to select resources by matching the knowledge points, and the learner's own characteristics are also used as constraints to obtain the final personalized learning sequence.

### 3.3.1. Matching Degree

The matching degree between knowledge points and resources refers to the selection of resources that are more closely related to the current knowledge points from the massive learning resources. The quantification of the matching degree in this paper takes into account not only the close correlation between current knowledge points and learning resources but also the coverage of knowledge points previously learned by the resources, which is more conducive to improving the learning efficiency of learners. Therefore, the specific calculation of the matching degree between knowledge points and resources in this paper is presented as follows:

$$M_r = |\frac{m_j}{s} - \sigma| \times \frac{cou_i}{\sum\limits_{k=1}^{m_j} cou_{jk}} \tag{18}$$

where $M_r$ represents the matching degree between knowledge points $i$ and resource $j$, $m_j$ is the number of knowledge points investigated by resource $j$, $s$ is the number of knowledge points mastered by learners, $\sigma$ is a regulating parameter, indicating the coverage of the resource on the knowledge points that learners have mastered, and its value is a constant between [0, 1]. Ideally, resources should cover all the knowledge points learned by learners. Therefore, the value of $\sigma$ is 1. $cou_i$ is the frequency of knowledge point $i$ in resource $j$, and $cou_{jk}$ is the total frequency of all knowledge points in resource $j$.

### 3.3.2. Adaptability of Learners and Resources

The fitness of learners and resources mainly refers to whether the recommended resources are suitable for the current learning level of learners. Different from traditional recommendation methods, today's online education environment can recommend suitable learning resources for different learners, that is, realize personalized recommendations. This paper quantifies fitness via learner characteristics to achieve personalized recommendations. According to the learner model constructed in this paper, the learner characteristics included in fitness quantification mainly include learning ability, learning style, and learning intensity. Learning ability mainly measures whether the difficulty of recommended resources is consistent with the current ability level of learners; learning style mainly selects appropriate resource types for learners; and learning intensity ensures that the learning time of the recommended resources is within the tolerance range of learners. Therefore, the specific calculation of the fitness of learners and resources is presented as follows:

$$G_f = \lambda C_i \times \frac{1}{1 + e^{-(|L_i - H_j|)(|S_i - D_j|)}} \tag{19}$$

where $G_f$ represents the matching degree between learner $i$ and resource $j$, $C_i$ represents the learning style of learners, $\lambda$ is the control parameter, specifically, the ratio of resource $j$ type to learning style type, namely, $\lambda = tr_j/t_i$, where $t_i$ indicates the resource type to which learners with learning style $C_i$ should correspond. The learning resources in this paper mainly include text, video, pictures, hypertext, animation, and tables. $tr_j$ is the type of current resource $j$. If $\lambda = 1$, resource $j$ is suitable for learner $i$,; otherwise, it is unsuitable, that is, $\lambda = 0$. These conditions refer to learning intensity. $H_j$ is the learning duration of resource $j$, $S_i$ is the learning ability, and $D_j$ is the difficulty of the resource, directly obtained from the junior high school mathematics knowledge base.

In summary, the objective function of personalized learning path recommendation is expressed as:

$$F = \omega_1 M_r + \omega_2 G_f \tag{20}$$

where $F$ represents the objective function, $\omega_1$ and $\omega_2$ are the weights, and $M_r$ and $G_f$ represent the matching degree and fitness, respectively.

## 4. Experiments and Results

In this paper, a personalized learning path recommendation model is designed by combining the association rule apriori algorithm and PSO algorithm. To verify the effectiveness and feasibility of the model in many aspects, the simulation experimental design and analysis of the experimental results will be described in detail.

### 4.1. Experimental Data

In this paper, data crawling technology is applied to obtain the log data of the online learning platform, and data processing technology is utilized to obtain the learning behaviour data of learners, including video viewing time, resource download, homework submission times, and other data. To improve the quality of the data, we need to use the program to further optimize the extracted learning data. Considering the complexity of the research problem, we use the method of random extraction to select learner data as the experimental object to verify the performance of the learning path recommendation model. The data support of the knowledge point sequence is 584 knowledge points from the junior high school mathematics knowledge map.

### 4.2. Experimental Design

The core of the recommendation model in this paper is divided into two parts: the first part is the generation of knowledge point sequences, and the second part is the recommendation of personalized learning paths based on the generated knowledge point sequences. Therefore, the entire experiment also focuses on these two parts to verify the performance of the model. The specific process is presented in the following section.

#### 4.2.1. Experiment of Knowledge Point Sequence

In this paper, the apriori algorithm is selected to generate a knowledge point sequence, which generates frequent item sets based on minsupport and minconfidence during execution. The difference in values will affect whether the knowledge point sequence can be accurately generated. The knowledge point sequence of this paper is 584 knowledge points for junior middle school mathematics, so to determine the minimum support degree and minimum confidence value, the accuracy serves as the evaluation standard. If the knowledge points k are set as 100, 250, 380, and 500, then the accuracy is optimal when the minimum support degree and minimum confidence degree are set. The accuracy is calculated as follows:

$$precision = \frac{\sum_{i=1}^{n} |R_i \cap T_i|}{\sum_{i=1}^{n} |R_i|} \tag{21}$$

where $R$ is the recommended result for learners, $T$ is the actual result of learners, and $n$ is the number of knowledge points.

The final knowledge point sequence is based on the knowledge graph, and the correlation between knowledge points is introduced to improve the closeness between knowledge points. Therefore, to verify whether the generated knowledge point sequence fully meets the needs of learners, this paper introduces coverage to evaluate the accuracy of the sequence:

$$coverage = \frac{\sum_{i}^{n} R_i}{n} \tag{22}$$

where $R$ is the recommended result for learners and $n$ is the number of knowledge points.

#### 4.2.2. Experiment of Recommended Model

To verify the effectiveness of the personalized learning path recommendation model proposed in this paper, this part of the experiment is designed from the following three aspects. First, the online learning platform learners are randomly divided into two groups, with 50 people in each group. One group chooses the learning sequence given by the personalized learning path recommendation model in this paper, which is referred to as the

experimental group, while the other group learns on the online learning platform according to their own needs, which is referred to as the nonexperimental group. The learning data of these two groups of learners is collected. The effectiveness of the recommendation model proposed in this paper is analysed from the two perspectives of learning outcome and learning time. Second, the stability of the PSO algorithm applied to the personalized learning path recommendation problem is analysed from four perspectives: average value (AVG_V), standard value (CHA_V), variance (VAR_V), and optimal solution (OSF_V). Last, the convergence rate of the PSO algorithm applied to the personalized learning path recommendation problem is compared with other algorithms to determine whether the fastest convergence rate is always maintained with different iterations. The contrast algorithms designed in the experiment are PSO, genetic algorithm (GA), and ant colony algorithm (ACO), which can be applied to solve the problem of learning path recommendation. However, their convergence effect will change when they cooperate with the apriori algorithm to solve this problem.

### 4.3. Result Analysis

After many simulation experiments, part of the experimental results are extracted to analyse the usefulness and feasibility of the personalized learning path recommendation model investigated in this paper.

In this paper, we use the apriori algorithm to generate the sequence of knowledge points, and in the apriori algorithm, the values of minsupport and minconfidence affect the accuracy of the algorithm, and the values of minsupport and minconfidence set too low will affect the accuracy of the algorithm, and set too high will cause the accuracy of the algorithm to decrease, therefore, in this paper, we design simulation experiments under different numbers of knowledge points to determine both according to the number of middle school mathematics knowledge points values. From Figure 2, the larger the values of minsupport and minconfidence, the higher the accuracy of the algorithm will increase. When minsupport varies between 0.4–0.6 and minconfidence varies between 0.8–1, the accuracy reaches the highest value, and then the accuracy tends to decrease when and continues to increase. Therefore, by experimental comparison, the values of minsupport and minconfidence are set to 0.53 and 0.91 in this paper.

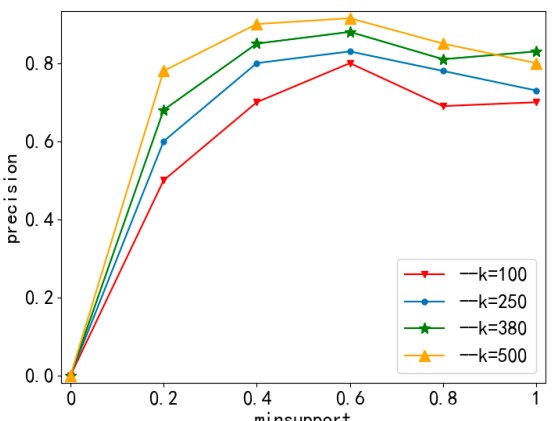 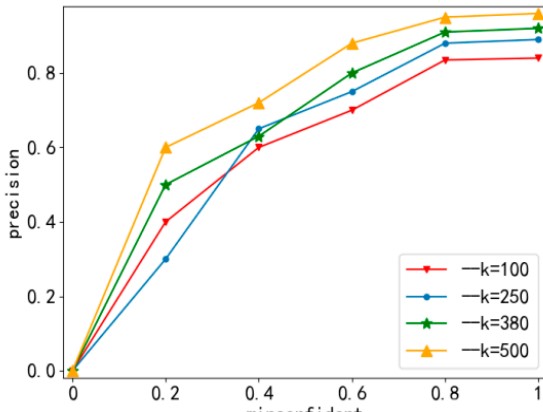

**Figure 2.** Parameter setting.

In this paper, the coverage of the generated knowledge point sequences is crucial for recommending personalized learning paths, since all knowledge points between learners' unmastered knowledge points and the target knowledge points are considered when generating the knowledge point sequences. In order to verify the coverage of the generated knowledge point sequences, this paper calculates the cognitive level of learners on the online learning platform based on the 584 knowledge points of junior high school mathematics, obtains the unmastered knowledge points of each learner, groups them into the

same set with the target knowledge points, and then generates knowledge point sequences based on the knowledge graph by the apriori algorithm and the correlation between the knowledge points, and calculates the learners' knowledge point. The experimental results from Table 4 show that the coverage of the knowledge point sequences generated for the learners by the method of this paper can reach more than 90%, which shows that the knowledge point sequences generated by the apriori algorithm and the correlation between knowledge points can more accurately generate the knowledge point sequences for the learners to meet their own needs.

**Table 4.** Coverage of knowledge points.

| Learner | NO-A | NO-B | NO-C | NO-D | NO-E | NO-F |
|---------|------|------|------|------|------|------|
| coverage (%) | 91 | 90 | 99 | 93 | 94 | 96 |

Based on the junior high school mathematics knowledge map, this paper uses the apriori algorithm to generate the sequence of knowledge points. To verify the feasibility of the model, this paper selects four different models to carry out comparative experiments with precision and coverage as indicators. These four models are the topological sorting algorithm (TO), firefly algorithm (FA), transfer learning method based on a knowledge map (TL-KG), and deep learning method (DL). During the experiment, the knowledge point data of 100 learners are selected and input into different models, and the output results of each model are recorded. In our simulation experiment results, the data of six randomly selected learners is shown in Table 5. According to Table 5, it can be seen that the apriori algorithm used in this paper outperforms the other four models in terms of precision and coverage, which indicates that the apriori algorithm is feasible and effective in generating sequences of knowledge points.

**Table 5.** Comparative experiment of the knowledge point generation algorithm.

| Learner | Target (%) | APRIORI | TO | FA | TL-KG | DL |
|---------|-----------|---------|------|------|-------|------|
| learner-A | precision | 88.2 | 72.1 | 83.6 | 87.3 | 84.2 |
|           | coverage  | 90.6 | 87.0 | 79.3 | 82.9 | 78.3 |
| learner-B | precision | 79.2 | 72.3 | 79.5 | 77.3 | 72.5 |
|           | coverage  | 77.8 | 69.3 | 75.3 | 71.8 | 74.7 |
| learner-C | precision | 97.0 | 95.2 | 94.3 | 92.1 | 85.4 |
|           | coverage  | 86.3 | 84.0 | 79.4 | 82.8 | 78.6 |
| learner-D | precision | 82.2 | 78.4 | 79.2 | 80.2 | 82.3 |
|           | coverage  | 93.1 | 80.3 | 86.4 | 90.1 | 85.2 |
| learner-E | precision | 75.5 | 70.4 | 72.0 | 69.8 | 74.1 |
|           | coverage  | 80.1 | 78.0 | 80.0 | 75.3 | 72.2 |
| learner-F | precision | 98.5 | 86.3 | 90.0 | 87.2 | 89.3 |
|           | coverage  | 89.3 | 85.0 | 82.9 | 87.4 | 88.0 |

The learners on the online learning platform were divided into an experimental group and a nonexperimental group, with 50 people in each group. Six learners were randomly selected from each group and analysed from the perspectives of academic performance and duration. Figure 3 shows that the learning duration of learners in the experimental group (recommended learning path) was less than that of learners in the nonexperimental group (not recommended learning path). In addition, the final learning performance is better than that of the nonexperimental group. Thus, when online learning platform learners learn according to the recommended learning path, the efficiency significantly improved. Therefore, it is necessary to recommend personalized learning paths for online learning platform learners.

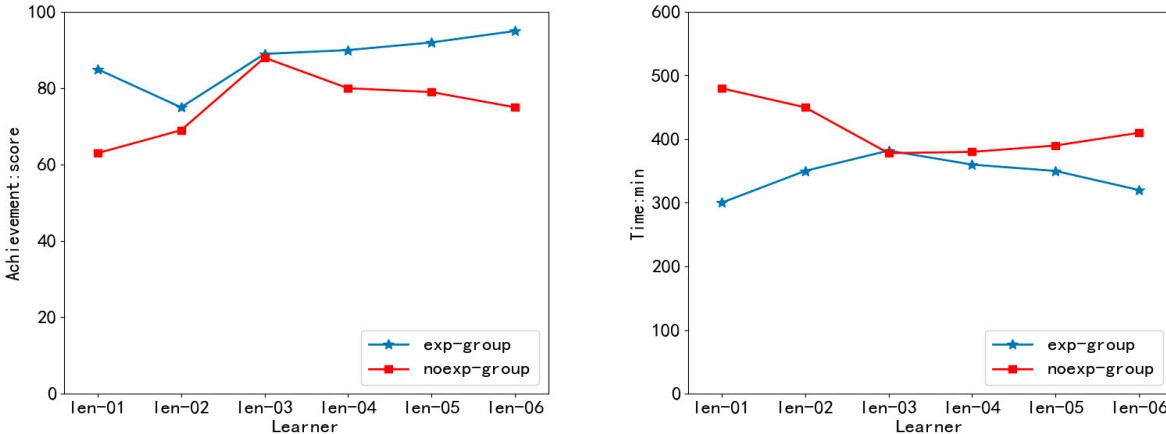

**Figure 3.** Comparison of learning results.

In this paper, the learning sequence is generated by PSO. To verify the stability of the algorithm applied to the personalized learning path, a comparison experiment is conducted with the traditional GA and ACO, and six learners are randomly selected. The AVG_V, CHA_V, VAR_V, and OSF_V were analysed. As shown in Figure 4, for the problem addressed in this paper, the PSO algorithm is obviously superior to the other two algorithms, and when solving the optimal solution, the stability is stronger than that of the other algorithms.

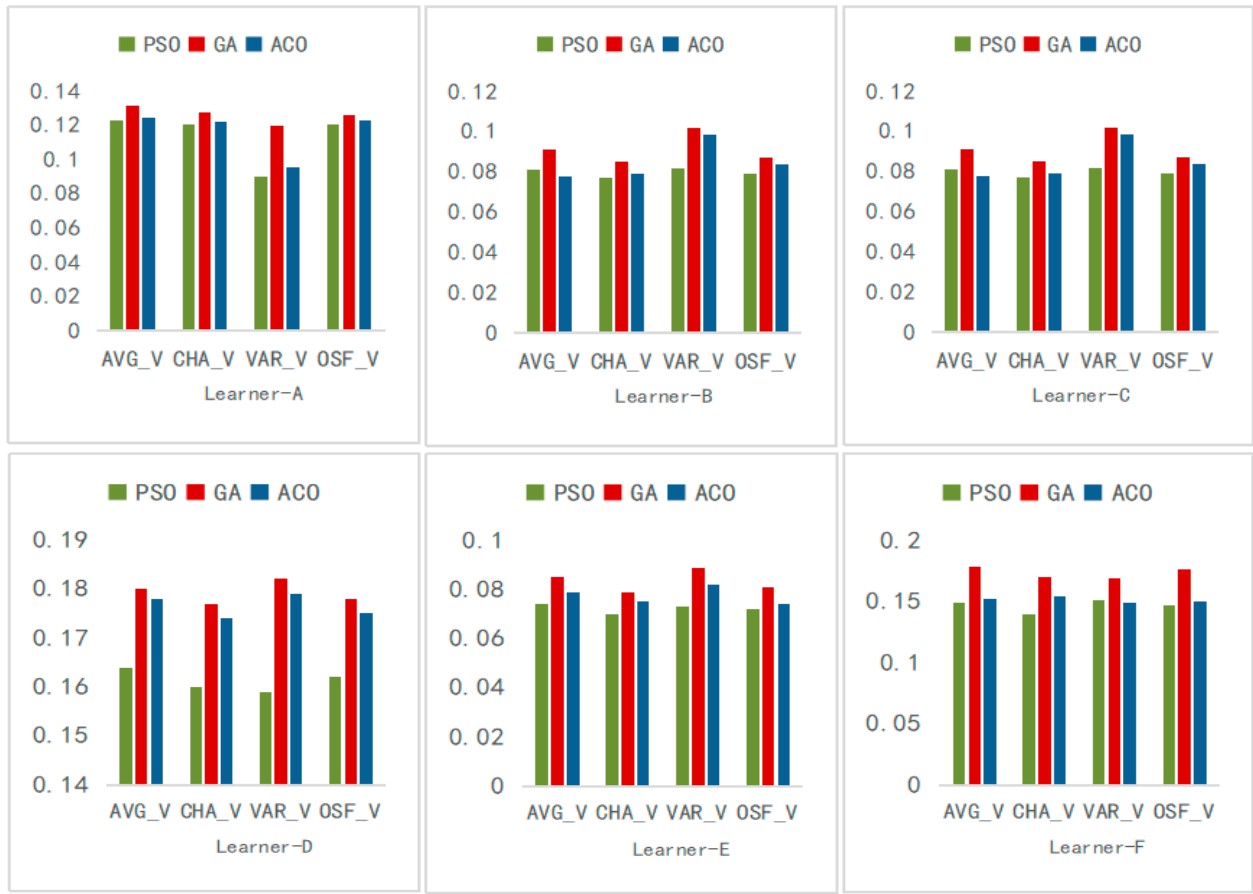

**Figure 4.** Comparison of the experimental results of the personalized learning sequence.

Assuming that to a certain extent, time can converge to infinity, the algorithm should not only find the optimal solution but also make the algorithm converge to a certain stable

value when searching for the global optimal solution to the problem, and its reference value is relatively low for the algorithm with poor convergence. In verifying the convergence speed of the particle swarm algorithm applied to the personalized learning path recommendation problem, some experimental data were randomly selected among 100 experimental results in this paper, and the iterative data of the algorithm were analyzed and the iteration graphs were drawn, as shown in Figure 5. The experimental results show that the convergence speed of PSO is significantly better than the traditional GA algorithm and ACO algorithm.

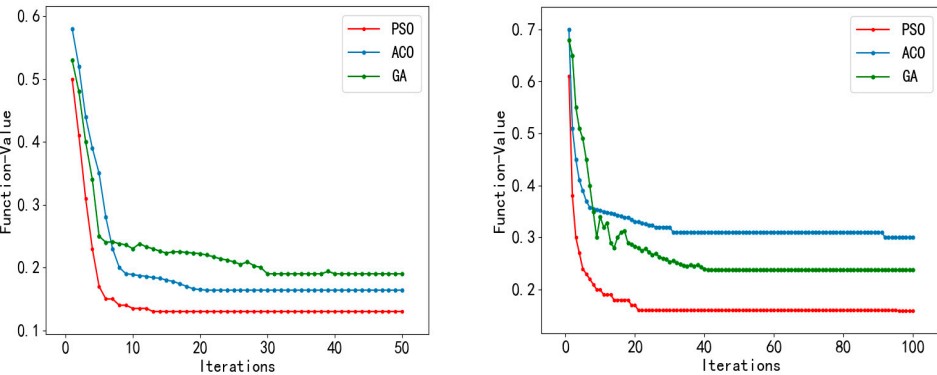

**Figure 5.** Comparison of algorithm iterations.

## 5. Conclusions

The rapid development of online education has introduced more high-quality learning resources to learners while causing the phenomenon that learners have lost their knowledge. Therefore, it is important to provide learning guidance for learners according to their own characteristics. This paper proposes a personalized learning path recommendation model based on multi-algorithm collaboration. First, based on the junior high school mathematics knowledge map, the apriori algorithm is used to generate a sequence of knowledge points for learners, and then the generated sequence of knowledge points is utilized as the learning order, integrating learners' cognitive level, learning ability, learning style, and learning intensity. Second, the PSO algorithm is selected to match personalized learning resources for each knowledge point to realize personalized learning path recommendations. Last, the effectiveness of the proposed model is verified by experiments. The experimental results show that compared with learners who do not use the recommended learning path, learners who used the recommended learning path in this model improved their efficiency in learning corresponding knowledge points, and their learning results significantly changed. This finding shows that for learners on the online platform, personalized learning paths can not only make them clarify the order of knowledge points but also help them learn these knowledge points according to the recommended resources to make learners' learning more targeted and efficient, which has significance for learners to better grasp knowledge and improve learning performance. This paper also compares the algorithms utilized in the model from different angles. The results show that the PSO algorithm is better than other algorithms for solving the problem of path recommendation in terms of stability and convergence and that when it cooperates with the apriori algorithm to recommend, the quality of the learning path is relatively high.

In future work, the learning path recommendation model can be optimized in the following aspects. (1) Expand the graph. This article takes the subject of middle school mathematics as the research object, and in future work, it can integrate multidisciplinary knowledge to provide learners with a more comprehensive learning path. (2) Delve into building a learner model. This article constructs a learner model from four perspectives. In future work, it can capture some hidden features such as emotional factors in learning to construct a more comprehensive and in-depth learning model. (3) Incorporate forgetting factors. Learners may forget a certain knowledge point over time, resulting in incomplete

mastery of some knowledge points. In future learning path recommendations, forgetting factors should be integrated to provide compensatory learning paths for some forgotten learned knowledge points, making it convenient for learners to promptly identify and fill in gaps, and improving learning efficiency. (4) Learning process detection. Learner learning is a dynamic, changing process; in future work, deep excavation of multi-dimensional characteristics of learners in the learning process, real-time detection of changes in characteristics such as cognitive level, learning ability, learning style and learning intensity of learners, and timely update of learners' status and learning paths.

**Author Contributions:** All the authors contributed equally to the conception of the paper. Y.M. proposed the methodology, paper organization, and experimental data, Y.M. prepared the original draft and validated the obtained results, L.W., F.L. and Q.J. supervised the project, L.W. and J.Z. revised and edited the final paper. All authors have read and agreed to the published version of the manuscript.

**Funding:** This work was supported by: (1) the National Natural Science Foundation of China under Grant 62176146; (2) the National Social Science Foundation of China under Grant 21XTY012; (3) the National Education Science Foundation of China under Grant BCA200083.

**Institutional Review Board Statement:** Not applicable.

**Informed Consent Statement:** Not applicable.

**Data Availability Statement:** The original data can be obtained by contacting the corresponding author.

**Conflicts of Interest:** The authors declare no conflict of interest.

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
