# Peer review of "A Personalized Learning Path Recommendation Method Incorporating Multi-Algorithm"

_applsci, doi:10.3390/app13105946_

Round 1

Reviewer 1 Report

Comments

1 – Most of the proposals in this paper are placed in section 2.1. I suggest to move the paper contributions to section 3.

2 – Equation (7) and line 169. Equation (7) has Tj and line 169 refers to it as Ti

3 – In Equation (7) use \alpha and 1 - \alpha instead of \alpha and \beta

4 – Please do not use symbols with double meaning.

                In equation (12), ‘n’ denotes the number of behaviour modes. Previously, in equation (9) it denotes the number of exercises

                In line 255, \miu denotes some mean value. In section 3.1, line 330, the same symbol denotes a cognitive level threshold

                In line 414, \eta is a control parameter. However, in equations (14) and (15) it has a different meaning.

                In equation (21) R is the recommended result for learners. However, in equation (17) R is the correlation.

5 – In equation (16), please define the c symbol

6 – Section 3.1. Please add a proper literature reference to the Apriori algorithm

7 – Equation (19) and line 420. Please explain how you compute the difficulty of the resource.

8 – Table 5. Please put the best results in boldface.

9 – Figure 3. Please state clearly the units of the metrics on the yy-axis.

10 – Figure 5. Please add more details on meaning of yy-axis regarding what is the “function_value”

11 – End of section 5. Please add some details on future work.

Comments on Writing

1 – The words “Research on” can be deleted from the title of the paper, without loosing value.

2 -  Line 42. Please do not use the “et al.” formulation, when referring to papers with less than 3 authors. “Wu et al. [1] “ should be “Wu and Fang [1]”

3 -  Line 49. “Son et al. [3] proposed proposes a “ -> “Son et al. [3] proposed a “

4 – Line 50. “recommender,this model can generate” -> “recommender. This model can generate”

5 – Line 54. “Wan et al. [5] proposed” -> “Li and Zhang [5] proposed“

6 – Line 64. “short-term memory(LSTM)” -> “short-term memory (LSTM)”

7 – At the end of section 1, please add a paragraph stating the organization of the remainder of the paper.

8 – Line 80. …the construction process of learner… -> …the construction process of the learner…

9 – Line 100. diagnostic model[8]. -> diagnostic model [8].

10 – Line 101. include the DINA[9] model and DINO[10] model -> include the DINA [9] and DINO [10] models

11 – Line 106. Fuzzy-CDF[11], -> Fuzzy-CDF [11],

12 – Line 114. skills[12].  -> skills [12].

13 – Please treat equations as elements of text. The punctuaction rules also apply to equations. For instance, at the end of equations (1), (2), and (3) there should be a comma…

14 – Line 139. “information and behaviour[13].”  -> “information and behaviour [13].”

15 -  Line 143. Researchers -> researches

16 – Line 165. considers two aspects of -> considers the aspects of

17 – Line 169. indicates to the learner's -> indicates the learner's

18 – Line 172. ?? and ?? indicates the number -> ??, and ?? indicate the number

19 – Lines 212 and 216. felder silverman -> Felder Silverman

20 – Lines 214 and 215.

information input: visual and verbal;

->

information input visual and verbal;

21 – Line 263 and 265. ?, ?, ? and ? are at -> ?, ?, ?, and ? are at

22 – Line 275. “related resources and organize them.” -> “related resources and to organize them.”

23 – Line 292. particle swarm optimization(PSO) -> particle swarm optimization (PSO)

24 – Line 297. learning style and learning intensity -> learning style, and learning intensity

25 – Line 307. Then, the association rule algorithm apriori -> Then, the aprori association rule algorithm

26 – Line 319. is used to explore k+1 item set. [16]. -> is used to explore k+1 item set [16].

27 – Line 327. is a set of learners -> contains learners

28 – Line 344. Those in the set ?? that  -> Those in the ?? set that

29 – Line 419. These conditions refers to learning intensity. -> These conditions refer to the learning intensity.

30 – Line 474. groups of learners are collected -> groups of learners is collected.

31 – Line 482. genetic algorithm(GA) -> genetic algorithm (GA)

32 – Line 483. ant colony algorithm(ACO) -> ant colony algorithm (ACO)

33 – Lines 511 and 518. COVERSION -> COVERAGE

34 – Line 516. learners are shown in Table 6. -> learners is shown in Table 6.

35 – Line 536. solved in this paper -> addressed in this paper

36 – References

Ref. 9

with few assumptions,and connections -> with few assumptions, and connections

Ref. 12

fuzzy control:a practical -> fuzzy control: a practical

Very good.

A few minor corrections are necessary.

Reviewer 2 Report

I am writing comments for the manuscript titled "Research on Personalized Learning Path Recommendation 2 Based on Multi-algorithm Collaboration" 
As per my evaluation few comments are:
1) In the related work, the disadvantages of the reviewed work should be mentioned.
 2) More recent references should be referred such as https://doi.org/10.1111/exsy.13247, https://doi.org/10.1007/s11277-022-097196, DOI: 10.1109/TCSS.2022.3200890, https://doi.org/10.1007/s11042-022-119727, https://doi.org/10.1016/j.cosrev.2021.100413
3) Result analysis the important parts of this paper lack a lot of details.
4) The text is not clear in writing and needs to be edited.

5) The Title seems to me a bit longer.

The English throughout the manuscript is weak and requires improvement.

Reviewer 3 Report

·         How did you select the values for the threshold of cognitive level ?, minimum support, and minimum confidence in the apriori algorithm? Are these values universal or do they need to be adjusted for different learning platforms or subjects?

·         How does the multi-algorithm collaborative personalized learning path recommendation model perform when compared to other existing learning path recommendation models? Are there any benchmarks or comparisons in terms of accuracy, efficiency, and personalization?

·         In the process of generating knowledge point sequences, how do you account for the differences in learners' initial understanding and knowledge of the subject matter?

·         Can the model adapt to changes in learners' cognitive level, learning ability, learning style, and learning intensity over time? If so, how does the model handle updates to these characteristics?

·         How does the model handle situations where a learner has difficulties in understanding the recommended learning path or fails to progress as expected? Is there any mechanism for providing alternative learning paths or additional support?

·         Is the proposed model suitable for all types of learners, including those with learning disabilities or other special needs? How can the model be adapted to cater to such diverse learner populations?

·         How does the model handle the inclusion of new learning resources or updates to the existing knowledge map? How can the model be kept up-to-date with the latest developments in a particular field of study?

·        Lines 285-286, “This paper describes the characteristics of learners from the three perspectives of cognitive level, learning ability, learning style and learning intensity” - The list should have four items, not three.

·         Lines 271-272: "The advent of educational informationization has promoted the rapid development 271 of the education industry." - Replace "informationization" with "informatization".

·        Lines 279-280: "First, the cognitive level of learners is obtained through the cognitive diagnosis model to obtain the knowledge point of learners." - Consider revising to: "First, the cognitive level of learners is determined using the cognitive diagnosis model to identify the knowledge points of learners."

Round 2

Reviewer 3 Report

Paper may be accepted in the current form. 

Paper may be accepted in the current form.